# Sodium Selenite Alleviates Breast Cancer-Related Lymphedema Independent of Antioxidant Defense System

**DOI:** 10.3390/nu11051021

**Published:** 2019-05-07

**Authors:** Hye Won Han, Eun Joo Yang, Seung-Min Lee

**Affiliations:** 1Department of Food and Nutrition, Brain Korea 21 PLUS Project, College of Human Ecology, Yonsei University, Seoul 03722, Korea; hyeewoon@naver.com; 2Department of Rehabilitation Medicine, Seoul National University Bundang Hospital, Seoul National University College of Medicine, Seongnam 13620, Korea

**Keywords:** breast cancer-related lymphedema, sodium selenite, supplementation, bioimpedance analysis, oxidative stress

## Abstract

Long-term surveillance is necessary to identify patients at risk of developing secondary lymphedema after breast cancer surgery. We assessed how sodium selenite supplementation would affect breast cancer-related lymphedema (BCRL) symptoms and parameters in association with antioxidant effects. A randomized, double-blind, controlled trial was conducted on 26 participants with clinical stage II to III BCRL. The control group (CTRL, *n* = 12) and selenium group (SE, *n* = 14) underwent five sessions of 0.9% saline and 500 μg sodium selenite (Selenase^®^) IV injections, respectively, within 2 weeks. All patients were educated on recommended behavior and self-administered manual lymphatic drainage. Clinical diagnosis on lymphedema by physicians, bioimpedance data, blood levels of oxidative markers, including glutathione (GSH), glutathione disulfide (GSSG), malondialdehyde (MDA), glutathione peroxidase activity (GSH-Px), and serum oxygen radical absorbance capacity (ORAC) levels, were investigated at timelines defined as baseline, 2-week, and follow-up. Sodium selenite increased whole blood selenium concentration in the SE group. Compared to the baseline, at 2 weeks, 75.0% of participants in clinical stage showed improvement, while there was no change in the CTRL group. At follow-up, 83.3% and 10.0% of the SE and CTRL, respectively, showed stage changes from III to II (*p* = 0.002). Extracellular water (ECW) ratios were significantly reduced at 2 weeks and follow-up, only in the SE group. Blood GSH, GSSG, GSH/GSSG ratio, MDA, and ORAC levels did not change by selenium supplementation. Sodium selenite improved diagnostic stages of BCRL along with ECW ratios, although the beneficial effect might not be related to its antioxidant activity. Selenite’s effect on lymphedema may be associated with non-antioxidant properties, such as anti-inflammation and immune function. Further mechanistic research using a larger population is needed.

## 1. Introduction

Breast cancer survivors are susceptible to the development of breast cancer-related lymphedema (BCRL), a chronic complication that results from surgical disruption of the lymphatic system [1]. The most obvious clinical manifestation of BCRL is swelling in the arm, shoulder, and neck of the affected site [2]. It can also result in disfiguration, impaired functional ability, physical discomfort, physiological distress, and development of other complications such as cellulitis and lymphangitis [3,4,5]. It is more common in women with high body mass index (BMI) values and women who have undergone extensive surgeries characterized as axillary lymph node dissection (ALND) and with a greater number of dissected lymph nodes [6]. A meta-analysis revealed that the estimated incidence rate of BCRL is 16.6%, which increases until 2 years after diagnosis or surgery [6]. Furthermore, in a prospective cohort study, BCRL occurrence was reported to increase progressively from 12, 30, and 60 months after surgery [7]. Studies on BCRL incidence found that the disease required long-term surveillance and maintenance [6,7]. Commonly used therapeutic methods include compression, exercise, pharmacological therapy, and manual lymphatic drainage (MLD) [3]. For pharmacological interventions, benzo-pyrones and selenium have been used in clinical trials [8].

Selenium is a trace mineral, found in selenoproteins, and it is of fundamental importance in human health [9]. Selenium plays an essential role in the antioxidant defense system, as it is incorporated into selenocysteine at the active site of multiple antioxidant selenoproteins, such as glutathione peroxidases (GSH-Px), thioredoxin reductase, and selenoprotein P [10]. Recently, the biological roles of selenium have expanded as being not only antioxidants but also immune-stimulating and redox-activating agents [11]. For instance, cancer cells show abundant sulfhydryl (thiol) expression [12], which induces disulfide exchange reaction between fibrinogen, forming a parafibrin coat around the cells to prevent immune detection [13]. Parafibrin is a protein that specifically coats tumor cells and to protect them from the immune response [13,14]. Sodium selenite was suggested to oxidize polythiols to disulfides, which may increase cancer cell exposure to the immune system [11,14]. Also, selenium exerts anti-cancer effects through its prooxidative properties on initiation and promotion in cancer cells [14]. Furthermore, selenite has been shown to directly activate natural killer (NK) cells [14,15].

Oral sodium selenite (Selenase^®^, 800 μg) supplementation was first reported to aid acutely induced lymphedema by subsiding the inflammatory response, edema, and pain within 10–15 min [16]. Later, others also reported significant reductions in secondary lymphedema in clinical studies with larger study populations [16,17,18,19,20,21]. In one study, the effect of oral sodium selenite was reported in a randomized, placebo-controlled, double-blind study with post-mastectomy BCRL patients who received complex decongestive physiotherapy (CDP), a conservative treatment program that combines several treatment approaches [8,16]. Selenium was effective in reducing edema in the arm, decreasing the skinfold index, and increasing the skinfold mobility compared to the placebo group who only received CDP [16]. Furthermore, selenium administration induced reduction in edema volume by up to 25% when compared with the unaffected limb during the treatment period [21]. These beneficial effects were suggested to be due to improvements in the levels of blood glutathione (GSH), glutathione disulfide (GSSG), and 4-hydroxynonenal (4-HNE) in the BCRL patients [16,21]. In addition, intracellular depletion of GSH may induce apoptosis, necrosis, and autophagy of cancer cells and thus increase sensitivity to cancer therapy [22]. However, since there was also a reduction found in the unaffected site, caution has been expressed regarding interpretation of these findings [16,18,21]. More recently, the impact of selenium was evaluated in 48 radiation-related secondary lymphedema patients, including 12 breast cancer patients who underwent axillary dissection [18]. Depending on the scoring system, 78.6–85.7% patients showed an improvement in clinical stage, even though the changes did not achieve statistical significance [18]. However, an important limitation of the study is that it was not placebo-controlled and lacked a mechanistic approach for evaluating the effect [18].

In this study, we hypothesized that administration of sodium selenite would alleviate the symptoms of chronic breast cancer-related lymphedema through antioxidant effects. We investigated the effect of sodium selenite supplementation on clinical stage II to III chronic BCRL patients. Changes in bioimpedance analysis (BIA) values and blood markers associated with the antioxidant defense system were assessed to understand the underlying mechanism.

## 2. Materials and Methods

### 2.1. Participants

In total, 34 volunteer BCRL patients were recruited at the Bundang Seoul University Hospital between June 2012 and March 2015. Female participants aged over 19 were considered eligible if they had unilateral clinical stage II to III BCRL. Patients with bilateral breast cancer, BMI < 20, BMI > 30, those taking selenium supplement, and those having hypersensitivity to the drug or other components in the drug were excluded. The study protocol was approved by the Bundang Seoul University Institutional Review Board (Institutional Review Board Approval Number 02-2012-062) and abided by the Declaration of Helsinki. All participants were asked to sign a written consent following a full explanation of the study.

### 2.2. Study Design and Intervention

In this double-blind study, participants assigned to the selenium intervention received 500 μg of sodium selenite (Selenase^®^, 10 mL, Boryung Pharmaceutical Co., Ltd., Seoul, Korea) dissolved in 50 mL of 0.9% normal saline via IV injection. Participants who received the control intervention were given an identical volume of 0.9% normal saline via IV injection. The interventions involved five sessions lasting 2 weeks from the first session or baseline. Precisely, the second, third, fourth, and fifth sessions were 3.2, 6.6, 9.8, and 13.3 days after the first session, respectively, on average and the intervention schedule for both groups were not significantly different (Appendix A). All patients were educated on recommended behavior and self-administered manual lymphatic drainage (MLD). Blood samples and BIA values were collected at three time points: before the first session (baseline), immediately after the last session (2-week), and one month from the last session (follow-up). Specifically, 2-week was defined as 1.8 ± 3.8 days after the last session, while follow-up was defined as 31.1 ± 8.8 days after the 2-week session.

### 2.3. Clinical Diagnosis of Lymphedema Stage

Patients were diagnosed based on the three-part lymphedema staging system established by the International Society of Lymphedema (ISL) [23]. Clinical diagnosis based on the ISL guideline was determined by physicians assessing the degree of fibrosis, pitting, cell proliferation (skin change), and swelling reduction with elevation of the affected limb [23]. Patients were diagnosed three times (baseline, 2-week, and follow-up).

### 2.4. Data Collection

Breast cancer- and treatment-related factors were obtained from Bundang Seoul University Hospital registry records. Clinical data collected included cancer location (left/right), surgery date and type (sentinel lymph node biopsy, axillary lymph node dissection, or both), number of excised lymph nodes, and treatment types (radiation therapy or chemotherapy). Pathological stages of breast tumors were determined by the 8th edition of AJCC (American Joint Committee on Cancer) cancer staging system, based on TNM (tumor/node/metastasis) classifications [24]. The bio-impedance values of all patients were assessed using a bioelectric impedance device (InBody 720^®^, Biospace, Seoul, Korea) at three time points. Data calculated from impedance values included extracellular water (ECW), total body water (TBW) ratio (affected-to-unaffected site), and 1 kHz, 5 kHz, and 50 kHz single frequency bioimpedance analysis (SFBIA) ratio (unaffected-to-affected site). Information on the detailed surgical methods for five patients (three in the CTRL and two in the SE) was unavailable due to data loss.

### 2.5. Blood Collection

Blood samples were collected following an overnight fast (at least 12 h). Venous blood specimens were collected in 3-mL ethylenediaminetetraacetate (EDTA)-coated and 5-mL serum-separating tubes (SST) to collect whole blood and serum samples, respectively. SST blood specimens were centrifuged at 3000 × g for 15 min to obtain serum. Samples were stored at −80 °C until analysis.

### 2.6. Whole Blood Selenium Level Measurement by ICP-MS

Whole blood selenium levels were analyzed using inductively coupled plasma-mass spectrometry (ICP-MS) (Agilent 7700, Palo Alto, CA, USA). In total, 78 samples were analyzed using 500 μL whole blood per sample. A modifier solution was prepared by diluting Triton X-100 (Merck, Darmstadt, Germany) and the yttrium standard solution (Merck, Darmstadt, Germany) with dH_2_O to final concentrations of 0.05% and 0.001%, respectively. This solution was then sonicated for 5 min. Washing (5% HNO_3_ in dH_2_O) and calibration standard (selenium dissolved in 1% HNO_3_ solution to 25, 100, 200, and 400 μg/L) solutions (Merck, Darmstadt, Germany) were also prepared. In separate 15-mL metal-free tubes, 100 μL of calibrator or sample was mixed with 100 μL of 1% HNO_3_ solution. After briefly mixing, 3 mL of modifier solution was added to all tubes followed by thorough vortexing.

### 2.7. Thiobarbituric Acid (TBA) Reactivity

Whole blood malondialdehyde (MDA) level was assayed according to Wang et al. with minor modifications [25]. Whole blood (0.1 mL) or MDA standard solution (0–50 μM) was mixed with 0.2 mL of thiobarbituric acid (TBA) reagent (15% trichloroacetic acid, 0.375% thiobarbituric acid, and 0.25 M HCl dissolved in dH_2_O). After boiling for 15 min, samples were centrifuged at 1000 × g for 10 min at room temperature. Absorbance of the supernatant was measured at 532 nm (maximum absorbance of TBA-MDA complex) and 453 nm to correct errors resulting from compound interference during lipid peroxidation [26]. Corrected absorbance values were obtained by subtracting 20% of the absorbance at 453 nm from the absorbance at 532 nm [26]. Various concentrations of MDA solution (0, 0.625, 1.25, 2.5, 5, 10, 25, and 50 μM) were measured to obtain a standard curve (*R*^2^ = 0.999–1.000), by which the whole blood MDA level was calculated.

### 2.8. Oxygen Radical Absorbance Capacity (ORAC) Assay

Serum was dissolved in 75 mM potassium phosphate buffer (pH 7.4) and deproteinized with perchloric acid (PCA) (final concentration 3%). After centrifugation (4 °C, 13,500 rpm, 20 min), the supernatant was diluted in potassium phosphate buffer. The Tecan GENios multi-functional plate reader (infinite F200, Salzburg, Austria) was used for the assay, with fluorescent filters at 485 nm (excitation) and 535 nm (emission). After adding the reactants, fluorescein and 2,2′-azo-bis (2-amidinopropane) dihydrochloride (AAPH) were added to a final concentration of 40 nM and 20 mM, respectively. The ORAC values were calculated from the net area under the fluorescence curve and peroxyl radical absorbance capacity (ORACROO•) was expressed as 1 μM Trolox equivalents (TE).

### 2.9. Glutathione Peroxidase (GSH-Px) Activity

Glutathione peroxidase activity was measured as described by Kim et al. [27]. Hemolysate (10 μL) was mixed with 100 μL 1 M Tris-HCl-5 mM EDTA buffer (pH 8.0), glutathione-reductase solution (10 U/mL), 2 mM NADPH, 20 μL 0.1 M glutathione solution, and dH_2_O up to a final volume of 1 mL. For the blank or control, hemolysate was replaced with an equal volume of water. After incubation (37 °C, 10 min), 10 μL of 7 mM *tert*-butyl hydroperoxide was added and the absorbance was immediately measured at 340 nm for 90 sec. NADPH disappearance was monitored by a decrease in A340 nm/min.

### 2.10. Analysis of Glutathione (GSH) and Glutathione Disulfide (GSSG)

Glutathione and glutathione disulfide levels in whole blood samples were measured by spectrometry using the GSH recycling method [28]. Total GSH (tGSH) and GSSG were measured and GSH was calculated (GSH = tGSH − GSSG). Whole blood samples used for GSSG analysis were pretreated with N-ethylmaleimide (NEM) to prevent auto-oxidation of GSH to GSSG. NEM-treated whole blood was mixed with equal volume of 15% TCA solution dissolved in water. After vortexing, samples were centrifuged (5 min, RT, 14,000 × g). Acid-deproteinized samples were mixed with 3-fold volumes of dichloromethane (DCM) and centrifuged (30 sec, 14,000 × g). The supernatant was then transferred to a new tube. For analysis, a 1.6-mL polystyrene cuvette (Sarstedt, Nümbrecht, Germany) was filled with PB200 (925 μL), 5,5′-dithiobis-(2-nitrobenzoic acid) (DTNB) (5 μL), sample (20 μL), and β-nicotinamide adenine dinucleotide 2′-phosphate (β-NADPH) (20 μL), in the order mentioned. Subsequently, 20 μL of glutathione reductase (GR; 20 IU mL^-1^) solution was added. Absorbance was measured at 412 nm for 1 min and repeated after the addition of 10 μL of 10 μM GSSG. For each sample, the concentration was normalized to the hemoglobin (Hb) level in the whole blood (nmol/g Hb). Total Hb level was measured using a diagnostic kit (Asan Pharm, Seoul, Korea) according to the manufacturer’s instructions.

Measurement of tGSH was conducted using whole blood samples without NEM; however, the deproteinization procedure remained unchanged. Samples were then diluted 1:100 with water. A cuvette was filled with PB200 (945 μL), DTNB (5 μL), sample (10 μL), and β-NADPH (20 μL), in the order stated. After adding 20 μL of GR solution, absorbance was measured at 412 nm for 1 min. The tGSH concentration was also normalized to Hb.

### 2.11. Statistical Analysis

Univariate statistical analysis was conducted using SPSS 24.0 (Statistical Package for the Social Sciences; SPSS Inc., Chicago, IL, USA). Comparison between the selenium group (SE) and control group (CTRL) was conducted using the Mann–Whitney test for continuous variables. Fisher’s exact test was performed for categorical variables with small expected numbers. A double-sided *p*-value < 0.05 was used as the criterion for significance. Effects of the interventions on continuous variables were analyzed using linear mixed-models (LMMs) with a first order autoregressive process (AR(1)) covariance structure, time as a repeated factor, and group, time, and time × group as fixed effects. In addition, subjects were added as random effects to account for dependence among observations for the same subject. Shapiro–Wilk testing for normality of residuals was also undertaken, confirming the suitability of a linear mixed effects model for modelling (*p* > 0.05). Effects of the interventions on lymphedema stage (ordinary data) were analyzed by generalized estimation of equations. Similar to LMMs, group, time, and time × group were added as fixed effects, and the subjects were added as a random effect.

## 3. Results

### 3.1. Participants

In total, 14 SE and 12 CTRL participants were analyzed in the study. The flow chart in Figure 1 summarizes the trial process. Demographic and clinical data for the SE and CTRL are presented in Table 1. There were no significant differences between the two groups in any demographic (age, BMI, and weight) or breast cancer-related characteristics (affected site, pathological stage group, recurrence, surgery method, number of dissected lymph nodes, time since first breast cancer surgery, and administration of radiotherapy and chemotherapy). Patients who received radiotherapy and chemotherapy were administrated with radiotherapy according to their clinical stage followed by neoadjuvant systemic chemotherapy (NAC) [29].

### 3.2. Effect of Sodium Selenite Supplementation on Blood Selenium Levels

Whole blood selenium levels were similar in both groups at baseline (CTRL: 164 ± 46 vs. SE: 164 ± 30 μg/L, *p* = 0.631). However, after supplementation, whole blood selenium concentration reached 215 ± 31 μg/L from 164 ± 30 μg/L in the SE group at 2 weeks (Figure 2). Increased selenium concentration was restored to 162 ± 33 μg/L at follow-up. Whole blood selenium concentration was significantly changed in the SE group (*p* < 0.001), but there was no significant difference in whole blood selenium concentration in the CTRL group during the study period (*p* = 0.499).

### 3.3. Effect of Sodium Selenite Supplementation on Lymphedema Clinical Stage

Immediately after the intervention, in 9 out of 12 stage III at baseline, the SE group demonstrated downstaging in clinical lymphedema stage III to II (Figure 3 and Appendix A). In contrast, in the CTRL group, no change in lymphedema stage was observed at 2 weeks (*n* = 0 out of 12) (Figure 3 and Appendix A). At follow-up, the SE group sustained the number of patients who underwent downstaging in lymphedema stage compared to baseline (*n* = 10 out of 14). However, the CTRL group did not show comparable changes in stage at the same time point (*n* = 1 out of 12). Overall, downstaging was significant in the SE group (*p* = 0.001) but not in CTRL group (*p* = 0.309).

### 3.4. Effect of Sodium Selenite Supplementation on Bio-Impedance Measures in BCRL Patients

BIA ratio values were unchanged by the intervention (Table 2). However, a time point comparison of ECW ratios showed significant reductions in the SE group, but not in the CTRL group. Compared to baseline, ECW ratios were significantly reduced both at 2 weeks and follow-up (*p* = 0.035 and *p* = 0.041, respectively) (Table 2).

### 3.5. Effect of Sodium Selenite Supplementation on Blood Parameters is Indicative of Oxidative Stress

Selenium intervention had no effect on whole blood GSH, GSSG, GSH/GSSG ratio, and serum ORAC values (Table 3). However, there were significant reductions in whole blood MDA in both the CTRL and SE groups (Table 3). There were significant decreases at the 2-week and follow-up time points in the CTRL group (*p* = 0.005 and *p* = 0.005, respectively), and there was a significant reduction at the follow-up time point only in the SE group (*p* = 0.040) (Table 3).

## 4. Discussion

In this study, intravenous sodium selenite administration (500 μg/day) for five sessions over 2 weeks showed beneficial effects on BCRL by improving lymphedema diagnostic stage. Most participants in the selenium-treated group, but only a few in the control group, presented reduced lymphedema, especially when the stage was III at baseline. However, the beneficial effect of selenium might not be related to its antioxidant activity.

Digestion of an inorganic form of selenium was associated with lower serum selenium concentration than selenomethionine, an organic form of selenium, suggesting less toxicity [30]. Excessive intake of selenium could lead to toxicity by increasing the serum selenium level acutely [30,31]. Previous clinical studies have used a range of 250 to 1000 μg/day of intravenous selenium and demonstrated no toxicity [32]. In another study, intravenous sodium selenite was tolerable up to 10.2 mg/m^2^ in patients with progressing cancer [33]. One study administered up to 4000 μg/day of selenium sodium selenite in severe septic shock patients [34]. The sodium selenite administration used in our study showed no severe adverse effects.

Our findings on the positive effect of selenium on lymphedema were consistent with previous reports. Previously, Zimmermann et al. reported that daily oral or intravenous sodium selenite (1000 μg/day) supplementation on the day of surgery and within 3 weeks post-surgery protected patients from oral tumor surgery-induced lymphedema [17]. The edema subsided more rapidly in the treatment group, as shown by a significant reduction in circumferential distance (tragus to tip of chin) [17]. Selenium administration as oral sodium selenite has been reported to significantly decrease edema volume and alleviate inflammation in breast cancer patients and head and neck cancer patients [16,35]. Kasseroller et al. reported significant improvements in edema volume along with other parameters such as skinfold index, skinfold mobility, subjective well-being, and incidence of erysipelas in a selenium-treated group compared with a placebo group [16]. In this placebo-controlled, double-blind study on 179 post-mastectomy BCRL patients, all patients received combined physical decongestive therapy; however, only the selenium-treated group received oral sodium selenite supplementation (1000 μg/day for one week, 300 μg/day for two weeks, 100 μg/day for 3 more months) [16]. Micke et al. reported a significantly positive effect of oral sodium selenite (500 μg/day for 4 to 6 weeks) in breast cancer and head and neck cancer patients who were diagnosed with radiation-induced secondary lymphedema [35].

In addition to the lymphedema stage scoring system, we used BIA to assess the effect of selenium supplementation on lymphedema. BIA measures impedance and resistance in multiple frequencies, which are used to calculate the amount of total body water and total body fluid (intracellularly and extracellularly) [36,37]. Others also used BIA as a prediction or early diagnosis method for lymphedema [38,39], and some studies have utilized this method to assess treatment effects [40]. In our study, only the ECW ratio was decreased in the SE group both at 2 weeks and follow-up. The BIA results were in accordance with the improvement in lymphedema stage and subjective assessment, indicative of potential benefits of selenium supplementation. Besides BIA, methods including circumferential distance, multiple lymphedema stage scoring systems, subjective index, skinfold index, and skinfold mobility were previously used to assess the alleviation of lymphedema symptoms [17,18,19,41].

The antioxidant properties of selenium have been implicated in human diseases such as cancer, atherosclerosis, and coronary heart disease [42,43]. Inorganic forms of selenium (selenite and selenate) have been suggested as being more effective in increasing platelet GSH-Px activity compared with organic (selenomethionine) forms [44,45]. However, there has been no experimental evidence on the connection between the antioxidant effects of selenium and its beneficial effects on lymphedema. In our study, the beneficial effects of selenium administration on lymphedema were not associated with the antioxidant system, including the activity of GSH-Px in the patients. Specifically, whole blood GSH, GSSG, GSH/GSSG, GSH-Px activity, and plasma ORAC showed no significant changes following selenium supplementation.

The lack of antioxidant effects of sodium selenite might have been related to the relatively high initial selenium levels in our study participants. In a previous report, the mean selenium concentration of lymphedema patients was 102.4 ± 19.8 μg/L, and 44% of lymphedema patients were considered selenium-deficient, which is below 100 μg/L (according to German authorities) [20]. However, in our study, the average baseline selenium level was far higher (164 ± 34 μg/L) than previous reports with similar stage BCRL patients (stage II, 107 ± 24 μg/L; stage III, 92 ± 14 μg/L) [16,20]. Our initial blood selenium concentration was much higher than previously reported whole blood selenium concentrations (89 μg/L or 1.13–1.15 μmol/L) [46], which might represent maximum plasma GSH-Px activity, thus resulting in an inability for further increases. In addition, the antioxidant effects of selenium supplementation appeared to depend on the initial selenium concentration in healthy human subjects [44]. With low doses of selenium supplementation (11–32 μg/day for 8–52 weeks) plasma and erythrocyte GSH-Px activities increased only if the initial plasma concentration was 20–60 μg/L [44]. Others reported no increase in plasma and erythrocyte GSH-Px activity following oral selenium supplementation (40–100 μg/day for 8.5–16 weeks) if the initial plasma selenium concentration was higher than 70 μg/L [44,47,48,49,50].

Possible explanations for the antioxidant-independent selenium effect of reducing lymphedema may include its properties related to the immune system and inflammation. In patients with primary or secondary leg lymphedema, Foldi et al. found that inflammation-associated gene expressions increased before and decreased after the first phase of treatment for lymphedema [51]. Sodium selenite has been introduced as a safe compound that may reduce lymphedema volume and act against inflammation [14,52]. Selenium-containing proteins suppress excessive immune response and chronic inflammation [20]. Dietary sodium selenite has been reported to increase lymphocyte proliferation in mouse models (2.00 ppm/day for 8 weeks) [53]. Oral sodium selenite supplementation (200 μg/day for 8 weeks) in healthy university students increased cytotoxic lymphocytes and NK cell activities [54]. Another study showed that the same treatment to healthy women did not show any change in the GSH-Px activity [55]. Selenium might have improved lymphedema symptoms through immune response pathways. Further studies are needed to unravel the selenium effect of lymphedema prevention with regards to immune system and inflammation.

Apart from lymphedema preventive effects, sodium selenite might have provided beneficial effects against cancer development. Selenite (Se^+4^) is more redox active than selenite (Se^+6^) due to its chemical reactivity differences [14]. As sodium selenite is transformed into elemental selenium in cancer, it oxidizes sulfhydryl groups [14], which then disrupts parafibrin, potentially increasing immune recognition towards cancer cells, thereby inducing cancer apoptosis [14,56]. Yu et al. reported a potential primary liver cancer-reducing effect of sodium selenite-supplemented salt with sodium selenite (providing 50 to 80 μg/day for 8 years) compared with normal salt [57]. The incidence increased when the selenium supplementation was removed [57].

This study is limited due to a small sample size in each group and insufficient data on the underlying mechanism of sodium selenite on lymphedema reduction. Our initial hypothesis of its antioxidant roles in BCRL was not supported, and due to insufficient remaining samples, further mechanistic studies could not be pursued.

## 5. Conclusions

In conclusion, our prospective study demonstrated the potential secondary lymphedema-alleviating effects of sodium selenite supplementation on BCRL patients. In our study, although intravenous sodium selenite supplementation did not show any significant antioxidant effects, it demonstrated immediate benefits on the clinical stages of lymphedema. It is possible that sodium selenite’s anti-inflammatory role, redox-active properties that increase immune sensitivity, and/or the activation of NK cells might have been related to the alleviation of lymphedema. Our findings support previous suggestions that sodium selenite supplementation could be a safe and cost-effective therapeutic for secondary lymphedema [18]. However, the exact mechanism of selenite on lymphedema prevention and/or treatment warrants further investigation.

## Figures and Tables

**Figure 1 nutrients-11-01021-f001:**
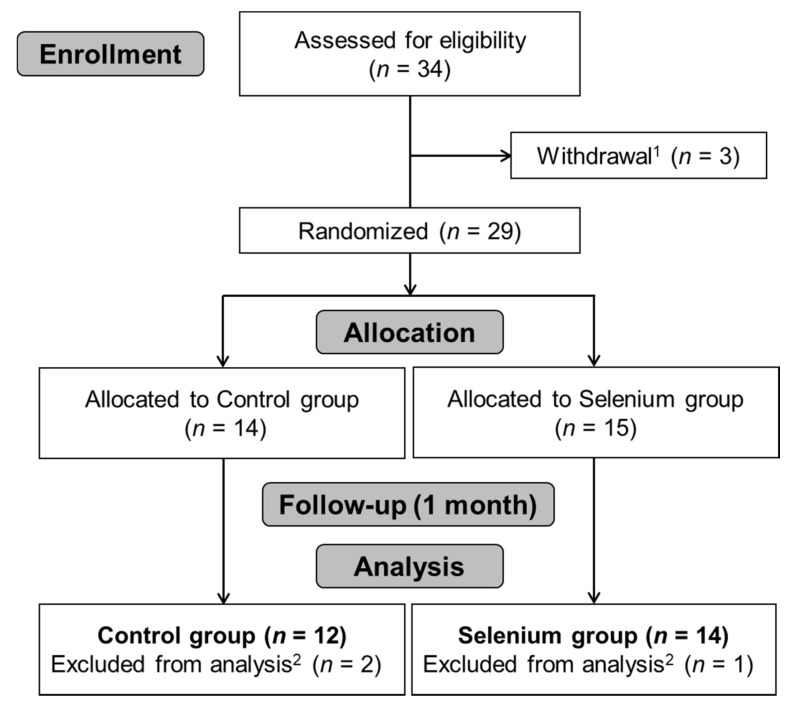
Flowchart describing progress during a randomized, placebo-controlled trial of selenium supplementation. ^1^ Three participants declined to participate in the study for personal reasons unrelated to the intervention (fracture, influenza, and transportation problems). ^2^ Outliers were excluded from the final analysis. CTRL: control; SE: selenium.

**Figure 2 nutrients-11-01021-f002:**
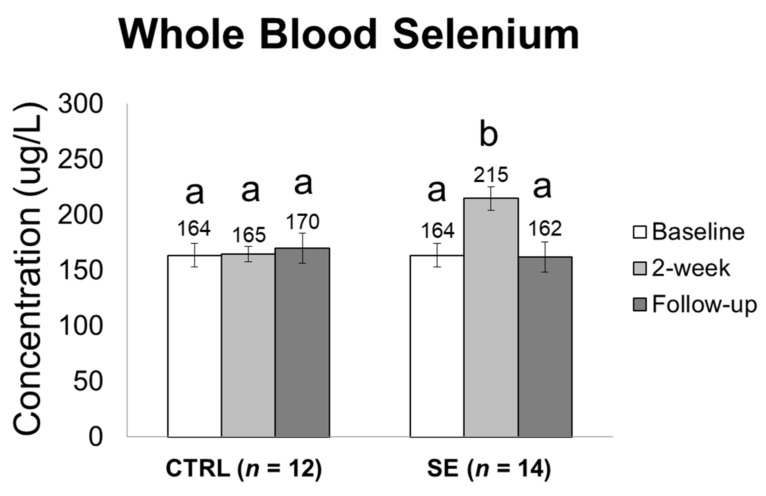
Selenium concentration in whole blood. Values are presented as mean ± standard error. Statistical significance was determined by linear mixed-models analyses using time as a fixed effect and subjects as a random effect. Bonferroni corrected post-hoc analysis was reported by linear mixed model using time, group, and time × group as fixed effects and subjects as random effects. Bars with a and b denote statistically significant differences (*p* < 0.001) whereas the same letters mean no significant difference.

**Figure 3 nutrients-11-01021-f003:**
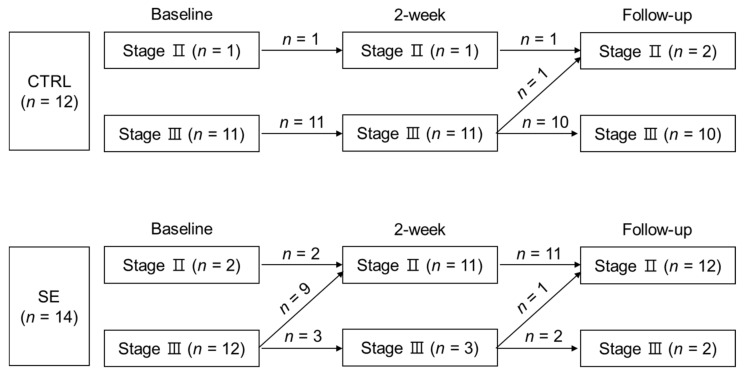
Changes in lymphedema stage during the study. Numbers in parentheses represent the number of participants at each stage. Numbers on the arrows indicate the number of participants who moved according to the direction of the arrow.

**Table 1 nutrients-11-01021-t001:** Demographic and surgical characteristics of participants.

	CTRL (*n* = 12)	SE (*n* = 14)	*p*
Age (years)	55.2 ± 13.9	48.0 ± 11.1	0.158 ^a^
BMI (kg/m^²^)	23.4 ± 2.3	25.2 ± 2.4	0.077 ^a^
Weight (kg)	59.0 ± 5.8	63.7 ± 7.6	0.145 ^a^
Affected site *n*, (%)			0.075
Left	10 (83.3)	7 (50.0)	
Right	2 (16.7)	7 (50.0)	
Pathological stage group *n*, (%)			0.416
II A	1 (8.3)	2 (14.3)	
II B	3 (25.0)	5 (35.7)	
III A	3 (25.0)	6 (42.9)	
III B	1 (8.3)	0 (0.0)	
III C	4 (33.3)	1 (7.1)	
Local recurrence *n*, (%)			0.449
Yes	2 (16.7)	1 (7.1)	
No	10 (83.3)	13 (92.9)	
Breast surgery method *n*, (%)			0.902
SLNB	1 (8.3)	2 (14.3)	
ALND	10 (83.3)	11 (78.6)	
Unknown	1 (8.3)	1 (7.1)	
Number of dissected lymph nodes	22.8 ± 9.1	24.8 ± 13.9	0.705 ^a^
Unknown *n*, (%)	3 (25.0)	1 (7.1)	
Post-surgery time *n*, (%)			0.914
1 < year	2 (16.7)	2 (14.3)	
1–3 years	5 (41.7)	5 (35.7)	
<3 years	5 (41.7)	7 (50.0)	
Radiation therapy *n*, (%)			1.000
Yes	12 (100)	12 (85.7)	
No	0 (0.0)	0 (0.0)	
Unknown		2 (14.3)	
Chemotherapy *n*, (%)			0.327
Yes	12 (100)	11 (78.6)	
No	0 (0.0)	1 (7.1)	
Unknown		2 (14.3)	

Values are presented as mean ± standard deviation (SD) or *n* (%). Unless noted otherwise, *p*-values were determined by Fisher’s exact test. ^a^
*p*-values were determined by Mann–Whitney test. Abbreviations: CTRL: control; SE: selenium; BMI: body mass index; SLNB: sentinel lymph node biopsy; ALND: axillary lymph node dissection.

**Table 2 nutrients-11-01021-t002:** Changes in bioelectrical impedance values during the study.

Parameters	Time Point	Time Point Comparison		*p*-Value	
		CTRL (*n* = 12)	SE (*n* = 14)	Baseline Comparison	Time × Group
TBW ratio	Baseline	1.31 ± 0.29	1.25 ± 0.20	0.705	0.129
	Δ 2-week	−0.04	−0.01		
	Δ Follow-up	−0.01	−0.04		
ECW ratio	Baseline	1.37 ± 0.32	1.29 ± 0.23	0.494	0.122
	Δ 2-week	−0.05	−0.03 *		
	Δ Follow-up	−0.01	−0.05 ^†^		
1 kHz SFBIA ratio	Baseline	1.40 ± 0.34	1.32 ± 0.25	0.595	0.307
	Δ 2-week	−0.04	−0.02		
	Δ Follow-up	−0.01	−0.04		
5 kHz SFBIA ratio	Baseline	1.40 ± 0.33	1.31 ± 0.25	0.595	0.123
	Δ 2-week	−0.06	−0.02		
	Δ Follow-up	−0.02	−0.04		
50 kHz SFBIA ratio	Baseline	1.36 ± 0.33	1.28 ± 0.23	0.860	0.129
	Δ 2-week	−0.04	−0.02		
	Δ Follow-up	−0.01	−0.03		

Values are presented as mean ± standard deviation for baseline. Changes from baseline to 2-week and baseline to follow-up are presented as Δ 2-week and Δ Follow-up, respectively. *p*-values for baseline comparison were derived from non-parametric Mann–Whitney *U*-test. *p*-values for changes in values were determined by linear mixed-models analyses using time, group, and time × group as fixed effects, subjects as random effects, and BMI as a covariate. *p*-value for time point comparison was obtained from Wilcoxon signed rank test (paired non-parametric t-test). * *p*-value < 0.05 in time point comparison between baseline and follow-up. ^†^
*p*-value < 0.05 in time point comparison between baseline and follow-up. Abbreviations: TBW: total body water; MFBIA: multiple frequency bioimpedance analysis; ECW: extracellular water; SFBIA: single frequency bioimpedance analysis.

**Table 3 nutrients-11-01021-t003:** Changes in oxidative stress parameter values following intervention.

Parameters	Time Point	Time Point Comparison		*p*-Value	
		CTRL (*n* = 12)	SE (*n* = 14)	Baseline Comparison	Time × Group
MDA (μM)	Baseline	12.0 ± 3.75	10.3 ± 3.94	0.212	0.115
	Δ 2-week	−2.97 **	−0.23		
	Δ Follow-up	−3.80 ^††^	−2.47 ^†^		
GSH (nmol/g Hb)	Baseline	6468 ± 2426	5751 ± 2437	0.462	0.869
	Δ 2-week	−172	−26.8		
	Δ Follow-up	+138	+600		
GSSG (nmol/g Hb)	Baseline	85.9 ± 16.5	85.4 ± 7.34	0.940	0.311
	Δ 2-week	−2.97	−0.23		
	Δ Follow-up	−3.80	−2.47		
GSH/GSSG ratio	Baseline	75.9 ± 29.5	68.8 ± 28.2	0.462	0.743
	Δ 2-week	−0.77	−3.81		
	Δ Follow-up	+5.10	+6.36		
GSH-Px activity	Baseline	72.1 ± 31.2	92.1 ± 38.0	0.860	0.742
	Δ 2-week	−4.00	+3.13		
	Δ Follow-up	−3.27	+2.55		
ORAC (TE)	Baseline	1025 ± 275	1027 ± 181	0.176	0.079
	Δ 2-week	+67.2	−40.9		
	Δ Follow-up	+26.6	−66.4		

Values are presented as mean ± standard deviation for baseline. Changes from baseline to 2-week and baseline to follow-up are presented as Δ 2-week and Δ Follow-up, respectively. *p*-values for baseline comparison were derived from non-parametric Mann–Whitney *U*-test. *p*-value for time point comparison was obtained from Wilcoxon signed rank test (paired non-parametric t-test). *p*-values were determined by linear mixed-models analyses using time, group, and time × group as fixed effects. ** *p* < 0.01 in time point comparison between baseline and 2-week. ^†^
*p* < 0.05, ^††^
*p* < 0.01 in time point comparison between baseline and follow-up. MDA: malondialdehyde; GSH: glutathione; GSSG: glutathione disulfide; GSH-Px: glutathione peroxidase; ORAC: oxygen radical absorbance capacity; TE: Trolox equivalents.

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
