# Peer review of "Sodium Selenite Alleviates Breast Cancer-Related Lymphedema Independent of Antioxidant Defense System"

_nutrients, 2019, doi:10.3390/nu11051021_

Round 1
Reviewer 1 Report
The article is very interesting.
The title should be expanded is definitely too short.
Signatures of the charts should be reduced. References should be checked.
Titles of articles should be written with a lowercase letter, example:29 (we write the names of journals with a capital letter and italics. The year of publication should be bold.
Check the requirements of the journal
However there are some information that should be clarified in the manuscript (major revisions) in order to be accepted for publication.
The aims of the study should be reformulated according to the research objectives and target results.
The conclusions must reflect the innovation of this study and the perspectives.
I would like to ask the authors to read the publication, Please add :
Application of sodium selenite in the prevention and treatment of cancers. Cells, 6(4), 39. (2017)
Pathophysiological significance of protein hydrophobic interactions: An emerging hypothesis. Medical hypotheses, 110, 15-22. (2018)
Please describe the properties of selenium as a compound with anti-cancer properties in introduction. Add these articles to discuss the results. Describing the chemical properties of sodium selenite on the development of cancer cells.
Please describe how selenium (inorganic) undergoes transformation in cancer cells.
What is the industrial use of the results obtained and perspectives for the future - please write a short commentary in the article referring to the food industry (Nutrients Journal).
How glutathione acts on cancer cells. The authors have not described this?
Please write. This will interest readers.
There is nothing about the dose of this chemical compound on the human body. Please describe.
Selenium in inorganic form also has toxic properties.
Author Response
Responses to Reviewer 1
Open Review
(x) I would not like to sign my review report
( ) I would like to sign my review report
English language and style
( ) Extensive editing of English language and style required
(x) Moderate English changes required
( ) English language and style are fine/minor spell check required
( ) I don't feel qualified to judge about the English language and style
Yes | Can be improved | Must be improved | Not applicable | |
Does the introduction provide sufficient background and include all relevant references? | ( ) | ( ) | (x) | ( ) |
Is the research design appropriate? | ( ) | (x) | ( ) | ( ) |
Are the methods adequately described? | ( ) | (x) | ( ) | ( ) |
Are the results clearly presented? | ( ) | (x) | ( ) | ( ) |
Are the conclusions supported by the results? | (x) | ( ) | ( ) | ( ) |
Comments and Suggestions for Authors
The article is very interesting.
(1) The title should be expanded is definitely too short.
Responses:
We expanded the title from “Sodium selenite alleviates breast cancer-related lymphedema” to “Sodium selenite alleviates breast cancer-related lymphedema independent of antioxidant defense system.”
(2) Signatures of the charts should be reduced.
Responses:
As the reviewer suggested, we have reduced excessive signatures and explanations in the charts (Fig. 1 and 2, Tables 1, 2, 3).
(3) References should be checked. Titles of articles should be written with a lowercase letter, example:29 (we write the names of journals with a capital letter and italics. The year of publication should be bold. Check the requirements of the journal
Responses:
We apologize not meeting the journal requirements regarding the references. We checked the references and revised the manuscript in the MDPI journal style.
(4) However, there are some information that should be clarified in the manuscript (major revisions) in order to be accepted for publication.
The aims of the study should be reformulated according to the research objectives and target results.
Responses:
As mentioned by the reviewer, we reformulated the aims of the study according to the research objectives and target results in the abstract (p.1, line 13 to 14). We included a statement (p. 2, line 77 to 78) in the Introduction explaining the aim of the study: “In this study, we hypothesized that administration of sodium selenite would alleviate the symptoms of secondary breast cancer-related lymphedema through antioxidant effects.”
(5) The conclusions must reflect the innovation of this study and the perspectives.
Responses:
To reflect the innovation and perspectives of the study, we added the following sentence the conclusion of the abstract: “Selenite’s effect on lymphedema may be associated with non-antioxidant properties such as anti-inflammation and immune function.” (p.1, line 29 to 30).
Also, we included some comments (p. 15, line 333 to 338) in the Discussion explaining the innovation that the selenite effect on lymphedema may not be due to antioxidant defense system but due to other methods related to inflammation, immune response, and redox activity.
(6) I would like to ask the authors to read the publication, please add:
l Application of sodium selenite in the prevention and treatment of cancers. Cells, 6(4), 39. (2017).
l Pathophysiological significance of protein hydrophobic interactions: An emerging hypothesis. Medical hypotheses, 110, 15-22. (2018)
(6-1) Please describe the properties of selenium as a compound with anti-cancer properties in introduction. (6-2) Add these articles to discuss the results. (6-3) Describing the chemical properties of sodium selenite on the development of cancer cells. (6-4) Please describe how selenium (inorganic) undergoes transformation in cancer cells.
Responses:
(6-1) We included the anticancer properties of selenium in the Introduction (p.2, line 50 to 57).
(6-2) As these thoughts are crucial to our hypothesis and conclusions, we added the suggested articles in the Discussion to discuss the results regarding the effects of sodium selenite as redox-active and immune-affecting compounds (p. 14, line 322 to 328).
(6-3) We described the chemical properties of sodium selenite, which clearly differs from selenite, on the development of cancer cells in the Discussion (p.14, line 323 to 324).
(6-4) We described how inorganic selenium is transformed in and affects cancer cells in the Introduction (p.14, line 324 to 326).
(7) What is the industrial use of the results obtained and perspectives for the future - please write a short commentary in the article referring to the food industry (Nutrients Journal)?
Responses:
As recommended by the reviewer, we added comments on the industrial use of selenium (sodium selenite) in the Discussion (p. 14, line 327 to 328; p.15, line 337 to 338).
(8) How glutathione acts on cancer cells. The authors have not described this? Please write. This will interest readers.
Responses:
We added the role of GSH on cancer in the Introduction: “GSH regulates cancer cell death, where intracellular depletion of GSH induces apoptosis, necrosis, and autophagy and increase sensitivity to cancer therapy” (p. 2, line 69 to 70).
(9) There is nothing about the dose of this chemical compound on the human body. Please describe. Selenium in inorganic form also has toxic properties.
Responses:
We have included the dose of sodium selenite based on previous clinical and experimental studies in the Discussion (p. 13, line 254 to 260).
Reviewer 2 Report
The authors present a randomized double-blind controlled Trial on sodium selenite in breast cancer-related lymphedema.
The paper is of interest and reflects the scope of the journal. The study is thorough.
However, grammar, language and style should be carefully revised.
The patients number is very limited, still the publication could be orthwile.
The authors should add more Information on the tumor stages of patients, i.e. T and N stage,
an more information on radiotherapy and chemotherapy should be added.
Author Response
Responses to Reviewer 2
Open Review
(x) I would not like to sign my review report
( ) I would like to sign my review report
English language and style
( ) Extensive editing of English language and style required
(x) Moderate English changes required
( ) English language and style are fine/minor spell check required
( ) I don't feel qualified to judge about the English language and style
Yes | Can be improved | Must be improved | Not applicable | |
Does the introduction provide sufficient background and include all relevant references? | (x) | ( ) | ( ) | ( ) |
Is the research design appropriate? | (x) | ( ) | ( ) | ( ) |
Are the methods adequately described? | ( ) | (x) | ( ) | ( ) |
Are the results clearly presented? | (x) | ( ) | ( ) | ( ) |
Are the conclusions supported by the results? | (x) | ( ) | ( ) | ( ) |
Comments and Suggestions for Authors
The authors present a randomized double-blind controlled Trial on sodium selenite in breast cancer-related lymphedema.
The paper is of interest and reflects the scope of the journal. The study is thorough.
(1) However, grammar, language and style should be carefully revised.
Responses:
We received full revision for grammar, language and style by a native English speaker (Editage).
(2) The patients number is very limited, still the publication could be worthwhile. The authors should add more Information on the tumor stages of patients, i.e. T and N stage, an more information on radiotherapy and chemotherapy should be added.
Responses:
We added the information on tumor stages of patients in Table 1 and Methods (p. 4, line 112-113): “Pathological stage group of breast tumor were determined by 8th edition of AJCC cancer staging system based on TNM classifications.”
Also, more information on radiotherapy and chemotherapy was added in the Methods (p.6, line 199 to 200).
Round 2
Reviewer 1 Report
The article has been corrected
I accept the publication